# Sexual Health among Youth in Residential Care in Spain: Knowledge, Attitudes and Behaviors

**DOI:** 10.3390/ijerph191912948

**Published:** 2022-10-10

**Authors:** Olga Fernández-García, María Dolores Gil-Llario, Rafael Ballester-Arnal

**Affiliations:** 1Department of Developmental and Educational Psychology, Faculty of Psychology, University of Valencia, 46010 Valencia, Spain; 2Department of Basic and Clinical Psychology and Psychobiology, Faculty of Health Sciences, Jaume I University, 12007 Castellón de la Plana, Spain

**Keywords:** sexual health, sexual knowledge, attitudes toward sexuality, sexual behaviors, sexism, condom use, adolescents, child welfare system, residential care, Spain

## Abstract

Adolescents in the child welfare system often face multiple maladaptive experiences that predispose them to worse sexual health outcomes. This study aims to (1) describe the sexual health of adolescents in Spanish residential care by exploring their sexual knowledge, attitudes toward sexuality, and sexual behaviors and (2) to find out whether there are certain characteristics that make a subgroup particularly vulnerable to engaging in risky sexual behaviors. A total of 346 adolescents recruited from 47 Spanish residential care facilities (34.1% girls, 65.9% boys) aged between 11 and 19 years old completed some self-report instruments. Descriptive analyses and tests to prove gender and age differences were conducted. Their knowledge of sexuality was lower than observed in the general adolescent population, their attitudes more negative, and their tendency to engage in risky sexual behaviors higher. Girls made very infrequent use of condoms, while boys had more sexist attitudes and made habitual use of withdrawal. Although more than 20% of them had experienced sexual exchange activities before the age of 13 until 17, they did not use condoms systematically. The low level of knowledge, the early initiation of sexual exchange activities, and the scarce use of protection methods, together with sexist attitudes, place this group in a situation of great vulnerability, increasing the risk of unwanted pregnancies, sexually transmitted infections, and even teen dating violence.

## 1. Introduction

Individual, family, and social experiences have a major impact on all areas of the individual and their sexual health outcomes [1]. Adolescents in the child welfare system who have had to be separated from their family and immediate environment to ensure their physical and emotional well-being are more likely to report histories of trauma or sexual abuse, living in poverty, having inconsistent, harsh, or unavailable parenting, and coming from families who experience multiple stressors [2,3,4]. For this reason, this vulnerable group of adolescents is at greater risk of experiencing unhealthy trajectories by sexual risk-taking, STDs, and teen pregnancy, among others, compared to adolescents not in the child welfare system [5,6]. 

In particular, these youths often lack a solid base of knowledge that would enable them to make adequate sexual health decisions [7]. Oman et al. [8], in their research with a sample of young people living in residential care facilities in different states in America, reported that their participants presented low knowledge about female anatomy and fertility and methods of protection. However, their knowledge was slightly higher on condom use and pregnancy risk, especially among girls [9]. Other studies claim that these young people have very little information about sexuality in general and condom use in particular or that the information they do have is erroneous [10]. Similar results were reported by Boustani et al. [11] regarding the HIV knowledge and safe sexual practices of a group of youth in residential care. It is believed that this lack of knowledge could be due, in part, to the frequent changes in residential care facilities and the problems of educational disruption experienced by these boys and girls. These situations make it difficult for them to receive information from the main educational contexts: their homes and their schools [7].

Lack of knowledge is directly related to the development of erroneous and negative attitudes about sexuality and the different aspects related to it. Oman et al. [8] stated that the attitudes of their study participants toward the use of contraceptive methods were positive; however, they reported that most of their sample agreed that condoms reduced pleasure, which calls into question their main conclusion. Furthermore, comparing their results with those obtained in studies of community samples, they added that adolescents in residential care, especially boys, have poorer attitudes than other adolescent groups, and that directed them toward sexual risk-taking. While half of the adolescents in the community sample indicated that they would be upset if they got pregnant or got someone pregnant, only a quarter of the youth in residential care would be upset. This positive or ambivalent attitude toward adolescent motherhood and parenthood, which is common in this group [11], would be behind the disproportionately high rate of early pregnancies. The desire for pregnancy is associated with family dysfunction, as it is perceived as a healing mechanism of childhood wounds or to make up for emotional shortcomings with their possible current baby or partner [12]. Likewise, as children learn and internalize observed family models, another consequence of growing up in potentially conflictual homes is the adoption of stereotypical gender roles and behaviors [13]. In this regard, some studies have found a high internalization of sexist attitudes among adolescents in residential care [14,15]. These beliefs are present through the idealization of the male role as caregiver and protector of women (benevolent sexism) and are more common in boys [14,16,17]. As in the youth in the community sample [18], hostile sexism, consisting of the expression of overtly negative beliefs toward females, is less common in these adolescents. However, age positively influences the expression of sexist attitudes in community adolescents [18,19], but this relationship has not always been found in adolescents in the child welfare system [14,20].

Thus, given the lack of appropriate knowledge and attitudes of these young people, it is not surprising that many of them have a higher likelihood of engaging in risky sexual behavior [21]. This vulnerable group of adolescents reports being sexually active at a younger age than adolescents of community samples [22,23]. Different studies with adolescents in residential care facilities in the Americas [24,25] postulate that, regardless of the participant’s gender, a remarkable percentage of their sample have had their first sexual activities at age 14 or earlier. The most frequent sexual activity appears to be vaginal intercourse together with oral sex, which is performed at an earlier age compared to anal sex [25]. The data from these studies contrast with those conducted with adolescents in the community sample. James et al. [23] found that while 6% of US high school youth have had their first sexual intercourse experience before age 13, this rate rises to 20% for youth in the child welfare system. 

Early sexual activity has been associated with a propensity to engage in additional risky behaviors, such as lower contraceptive use [26]. The lack of neuro-development of high-order cognitive skills that characterize individuals in early adolescence [27] leads to those who engage in sexual activities at a younger age being more likely to make poor decisions without thinking through the consequences of, for example, having unprotected sex [28]. Studies with youth in the child welfare system, in addition to reporting extremely low rates of contraceptive use in these adolescents [29], all agree on the significant influence of gender in the use of contraceptive methods, being in favor of boys, especially with respect to condom use [8,25,30]. Age seems to lead to greater use of contraceptive methods too. Research conducted with emancipation or out-of-care adolescents over the age limit reported higher rates of contraceptive use that approached those of their peers in the community sample [31,32]. This would confirm the hypothesis that a lack of cognitive maturation directly influences risk-taking and, thus, contraceptive use. However, residential care youth also cite finding it difficult to access condoms and general sexual health care [33], which would explain the differences found between adolescents in the child welfare system and their peers in a community sample. 

The lack of information on the sexual health of adolescents in the child welfare system in Spain and the need to provide these adolescents with interventions that would truly ensure their healthy sexual development led us to propose the following research. The first aim of this study is to describe the sexual health of adolescents in residential care in Eastern Spain by exploring their sexual knowledge, attitudes, and behaviors. The second objective is to find out whether there are certain characteristics of the sample (gender and age) that make a subgroup of these adolescents particularly vulnerable to developing poorer sexual health. As a consequence of these two objectives, the present study is not only to shed light on their sexual information and their sexual behavior but also allows to detect their main strengths and weaknesses in sexual education in order to develop educational interventions fully adapted to the needs of this high-risk population in Spain.

## 2. Materials and Methods

### 2.1. Data Collection

This cross-sectional study was conducted between June 2020 and May 2021, and 47 residential care facilities in Eastern Spain were recruited. A total of 346 adolescents in the child welfare system completed the Child and Adolescent Protection System Form, the Sexual Knowledge and Attitudes Questionnaire, the Contraception Methods Knowledge Questionnaire, the Sexuality and Health Knowledge Questionnaire, the Ambivalent Sexism Inventory, and the AIDS Prevention Questionnaire.

The inclusion criteria established for the selection of the sample were: (1) being 11 years of age or older (Finkelhor et al. [34] suggest that youth can respond to surveys with reliable information after age 10) and (2) living in a residential care facility at the time of the interview. The exclusion criteria were: (1) poor understanding of the Spanish language and (2) insufficient cognitive ability to understand and respond to the measurement instruments (information reported by residential care facilities professionals).

#### Procedure

Permission was obtained from the Directorate General for Childhood and Adolescence (DGCA) of the Valencian Region, the administrative agency responsible for implementing steps to protect minors and who have guardianship over them. Subsequently, the coordinators and directors of the residential care facilities were contacted to explain to them the project and to request authorization from them to enter the centers. These homes and residences offer comprehensive and educational care to children and adolescents in care and/or guardianship to those who are deprived of a suitable family environment. Once this first contact had been established, the center’s staff explained the proposal to the boys and girls of the residential care facility, and appointments were arranged with the minors who wished to participate on a voluntary basis. No financial assistance or compensation was offered to participants.

At the same time, three professionals from the research team with extensive experience in the evaluation and treatment of minors were trained in the application of this battery of instruments to avoid or reduce as much as possible the interviewer bias in the evaluation of the participants and ensure the maximum reliability and validity of the data collected. The application of the standardized instruments was carried out in person, individually, and in a comfortable environment of maximum privacy. Only the participant and one of the evaluators were present while the assessment was being completed. All participants were informed of the confidentiality of their responses before beginning to complete the battery, as well as their right to leave the study at any time. All residential care facilities and adolescents who chose to participate gave their prior authorization to do so. CAPSys had to be completed by a professional from the residential care facility (director, psychologist, educator, etc.), who knew the adolescent well and had been previously instructed to ensure full understanding of the items on the form. Participants always had the support of the expert who was available to resolve their doubts.

### 2.2. Measures Tool

#### 2.2.1. Child and Adolescent Protection System Form (CAPSys [35])

This instrument, validated in Spanish [35], consists of 67 items grouped into six dimensions: “*General information*” (9 items), “*School/work situation*” (9 items), “*Protection system history*” (9 items), “*Family visitation history*” (9 items), “*Biological family information*” (24 items), and “*Experiences of sexual abuse/maltreatment*” (7 items). For this study, only “*General information*”, which explores basic information about the adolescent (e.g., gender, date of birth, nationality, etc.), was used. It is an instrument designed to be completed by a professional from the residential care facility who knows the child well and has access to his or her record. The internal consistency of this instrument in this study ranged between 0.61 and 0.78.

#### 2.2.2. Sexual Knowledge and Attitudes Questionnaire (CAS [36])

The CAS consists of 34 dichotomous response items (agree/disagree) divided into two subscales: “*Sexual Knowledge*” (17 items; e.g., “Masturbation does not generate physical disorders”) and “*Sexual Attitudes*” (17 items; e.g., “Homosexuals are sick and vicious people”). This questionnaire was validated in Spanish [36]. Each subscale has a score range from 0 to 17. The internal consistency was acceptable for both subscales (α = 0.66 and α = 0.62, respectively) in this study.

#### 2.2.3. Contraception Methods Knowledge Questionnaire (ANTI [36])

This Spanish-validated self-reported instrument, consisting of 9 items, assesses knowledge about different contraception methods (e.g., “Withdrawal and natural methods are the best and most effective contraceptive methods”). The total score is obtained by adding the scores obtained in each of the items (0, disagree; 1, agree), and the range is between 0 and 9. The reliability analysis in this study was found as an α of 0.60.

#### 2.2.4. Sexuality and Health Knowledge Questionnaire (SYS [36])

The SYS is a Spanish-validated self-reported instrument to assess the level of knowledge about sexually transmitted diseases, and more specifically, HIV and its transmission (e.g., “A person carrying AIDS can transmit the infection even without having symptoms”). It includes 9 items, and the total score is obtained by adding all the items, where each item can be scored as 1 (agree) and 0 (disagree), and the score range is between 0 and 9. The reliability analysis in this study report an acceptable internal consistency (α = 0.62).

#### 2.2.5. Ambivalent Sexism Inventory (ASI [37,38])

The ASI, validated in Spanish [38], assesses ambivalent sexist beliefs about women. It comprises two subscales of 10 items, each measuring “*Hostile sexism*” and “*Benevolent sexism*”. Hostile sexism evaluates openly discriminatory attitudes and behaviors based on the supposed inferiority of women (e.g., “Boys should exert control over who their girlfriends interact with”). Benevolent sexism explores attitudes that even when expressed in a positive affective tone, stereotype and limit women to traditional roles (e.g., “Boys should take care of girls”). The sum of all of the scale items gives the total scale score for “*Ambivalent Sexism*” as the convergence of apparently positive (benevolent sexism) and negative (hostile sexism) attitudes towards women. All items are statements to which participants respond on a scale from 0 (“strongly disagree”) to 5 (“strongly agree”). Each subscale score ranges from 10 to 60, and the total scale score ranges from 20 to 120. The internal consistency was α = 0.91 for the total scale, α = 0.87 for the hostile sexism subscale, and α = 0.86 for the benevolent sexism subscale in this study.

#### 2.2.6. AIDS Prevention Questionnaire (CPS [39])

The CPS is a Spanish-validated [39] self-administered measure that includes 44 different response format questions. The main components are information and knowledge about HIV (12 items), attitudes and perceived self-efficacy (14 items), condom use intentions (6 items), safe sexual behavior (7 items), and, finally, stigma and discrimination towards people living with HIV (5 items). For this study, only the “Safe sexual behavior” component was used to explore sexual practices and the frequency of use of some contraceptive methods. The reliability analysis found a good internal consistency for this component in the study (α = 0.67).

### 2.3. Statistical Analysis

A descriptive analysis was conducted, including means (M) and standard deviations (SDs) for numerical variables (General knowledge, Contraception methods knowledge, sexually transmitted infections (STIs) knowledge, General attitudes, Sexism, and Age of first time of sexual practices), and frequencies (%), and the number of subjects (*n*) for categorical variables (Sexual practices, Contraception methods use, and Age groups of first time of sexual practices). 

The distribution of each of the dependent variables was examined for normality assumptions and compared by gender as well as by age. These two variables have been shown to be determinants in samples of community adolescents. Numerical dependent variables were related to age using Pearson correlation (r) and independent sample *t*-test to compare gender groups. Effect sizes (ESs) of the raw mean differences were obtained using the standardized mean difference, d [40]. For categorical dependent variables, analysis of variance (ANOVA), chi-squared, or independent sample *t*-test was performed. Further, odds ratios (OR) or standardized mean difference (d) were added to report the power of the association whenever possible. Age groups were also created to examine the trends according to age ranges following the stages of adolescence (early, middle, and late). 

Enough evidence to reject the null hypothesis was considered if *p* < 0.05. Following Cohen’s classification, the magnitude of the standardized d value can be interpreted as 0.25, 0.5, and 0.8 for small, median, and large effects on the outcomes of interest. ESs were calculated using an effect size calculator [41]. OR was considered statistically significant when its 95% confidence interval (CI) did not include the value 1. Data analyses were conducted using SPSS 26.

### 2.4. Ethic Considerations

The study complied with the ethical principles of the 1964 Declaration of Helsinki and was approved by the Experimental Research Ethics Committee of the University of Valencia (Spain).

## 3. Results

### 3.1. Sociodemographic Characteristics

Of the total participants, 34.1% (*n* = 118) were females compared to 65.9% (*n* = 228) who were males. The study sample was between 11 and 19 years of age (M = 15.73; SD = 1.76). Almost half of the adolescents (48.5%, *n* = 167) were between 14 and 16 years old, followed by 39.2% (*n* = 135) between 17 and 19 years old and only 12.2% (*n* = 42) who were between 11 and 13 years old. Regarding their sexual orientation, almost 85% identified themselves as heterosexuals, while 7.1% as bisexuals, 2.1% as homosexuals, and 0.3% as pansexual. Around 6% of the sample defined themselves as being in the process of exploring their sexual identity. Although the majority of the participants (57.9%) were born in Spain, there were 29.7% who were born in Morrocco, and the remaining nationalities were underrepresented (Eastern European: 4.7%; West African: 3.3%; South/Central American: 2.7%; Pakistani: 1.2%; and Portuguese: 0.6%). Among the adolescents 29.9% presented mental health problems (*n* = 100), and 33.3% had substance use problems (*n* = 115).

### 3.2. Sexual Knowledge

The total sample had a mean score close to the median of the range on the three scales assessing knowledge (Table 1). However, participants had more knowledge about STIs than contraception methods. The results also showed significant gender differences in knowledge about general sexuality (t = 5.47, *p* < 0.001), STIs (t = 3.89, *p* < 0.001), and contraception methods (t = 3.25, *p* = 0.001), favoring women. General sexual knowledge had a large ES for females (d = 0.62, 95% CI [0.39, 0.85]), contraception methods knowledge had a modest ES (d = 0.37, 95% CI [0.14, 0.59]), and STIs knowledge a medium ES (d = 0.45, 95% CI [0.22, 0.67]). Regarding age, a correlation was statistically significant with contraception methods knowledge (r = 0.2; *p* < 0.001). Participants between 14 and 16 years of age seemed to be the most knowledgeable about sexuality in general and STIs.

### 3.3. Sexual Attitudes

In the different attitudinal aspects evaluated, the influence of gender was significant. Women scored significantly higher on attitudes toward sexuality in general than men (t = 5.47, *p* < 0.001), and significantly lower on sexism which includes ambivalent, hostile, and benevolent (t = −7.88, *p* < 0.001; t = −6.8, *p* < 0.001; t = −7.44, *p* < 0.001; respectively). Ambivalent, hostile, and benevolent sexism had a large effect size for men (d = −0.77, 95% CI [−1.01, −0.54]; d = −0.85, 95% CI [1.08, −0.62]; d = −0.89 95% CI [−1.13, −0.66]; respectively). Regarding age, the correlations were not significant with any attitudinal aspects evaluated, except with hostile sexism (r = 0.14; *p* = 0.012). For the remaining variables, adolescents aged 14 and 16 years scored slightly higher on positive attitudes towards sexuality in general, younger adolescents scored slightly higher on benevolent sexist attitudes, and older adolescents scored slightly higher on ambivalent sexist attitudes. Table 1 shows these results.

### 3.4. Sexual Behaviors

Among 90% of adolescents in the sample had engaged in any of the sexual activities that they were asked about. More than 80% of the total sample had masturbated and almost 60% had masturbated as a couple. More than 70% had performed vaginal penetration and almost 50% had performed oral sex. The percentage of adolescents who had engaged in anal penetration was lower (18.8%) (Table 2).

The prevalence of sexual practices was not influenced by gender, with the exception of masturbation (χ^2^ = 51.44, *p* < 0.001), in which men had a significantly higher prevalence. No statistically significant differences were found between men and women with respect to the age at which they first engaged in each sexual practice. Regarding age, there were statistically significant differences in the mean age of adolescents who engaged in the assessed sexual practices compared to those who had not. The age of the participant at the time of assessment correlated positively and statistically significantly with the age of the first time of performance of each sexual practice (Table 2).

In addition, almost half (46.4%) of them had their first sexual activity before the age of 13, while 50.7% had it between the ages of 13 and 15 years and 2.9% after the age of 15 years. As shown in Table 3, masturbation was the earliest onset sexual practice, with 44% of our sample first engaging in it at 12 years of age or younger. Anal sex and oral sex were the sexual practices with the latest onset (25.4% and 23.8% performed for the first time at 16 years of age or older, respectively). 

Table 3 also shows the frequency of men and women having sexual activities at 12 or younger, between 13 and 15, or at 16 or older. 

As shown in Table 4, 42% of the sample used male condoms regularly, and 35% of them never used male condoms. The remaining contraceptive methods had a lower prevalence of use, with withdrawal being the next most prevalent (11.2%) after the male condom. The frequency of use of most contraceptive methods was influenced by gender (male condom, χ^2^ = 9.65, *p* = 0.008; withdrawal, χ^2^ = 6.99, *p* = 0.03; oral contraceptive pills, χ^2^ = 14.49, *p* = 0.001; hormone patches and injections, χ^2^ = 17.34, *p* < 0.001), with the exception of female condom use. Male condom use and withdrawal were in favor of men (43.5% and 13.9% for boys vs. 39.1% and 6.1% for girls, respectively), while oral contraceptive pills and hormone patches and injections favored women (4.2% and 2.3% for boys vs. 15.7% and 13.9% for girls, respectively). The differences with respect to the mean age of participants who used male condoms never, sometimes, and usually were statistically significant (F = 25.74, *p* < 0.001) as well as for hormone patches and injections use (F = 3.14, *p* = 0.045).

## 4. Discussion

The development of successful sexual health involves multiple domains, including the acquisition of positive skills and understandings [42]. There is absolute agreement in the literature on the importance of possessing certain knowledge about sexual development and sexuality that demystifies many of society’s strongly held beliefs [43], helps develop a positive and respectful attitude towards one’s own and others’ sexuality [44], and aids in the understanding that sexual behavior is, in and of itself, positive as long as the risks associated with certain sexual decisions are known and avoided [45]. This study aimed to explore the knowledge, attitudes, and behaviors of adolescents living in residential care facilities in Eastern Spain. 

Contrasting the sexual knowledge of adolescents in our sample compared to the sexual knowledge of adolescents in a community sample from the same country who were assessed with the same instrument [36], we can conclude that adolescents in residential care in Eastern Spain have poor knowledge about sexuality in general, including STIs and contraceptive methods. This could be due not only to the fact that this group receives much less sexual education because of their habitual mobility of facilities and worse access to sexual health services [33] but also because, given their earlier sexual activity, information about sexuality comes too late [12]. Other research on youth in the child welfare system also reported similar results [8,11]. In addition, our findings agreed with those of Claramunt [36], who reported that young people have more knowledge about STIs than about contraceptive methods or general aspects of sexuality (sexual practices, sexual response, etc.). This could be because STIs are also part of the compulsory school curriculum and, therefore, their learning is reinforced in the classroom. However, despite the over information that may be available to adolescents about STIs, there is an increase in them, probably favored by the lack of effective campaigns and educational programs against STIs.

Gender differences in sexuality have long been a research target. In our study, as in other literature with adolescents in the child welfare system in other countries (e.g., Combs et al. [9]), girls had more knowledge about sexuality. These results could be explained by girls’ more frequent access to sexual health services, a place to obtain information about human sexuality. However, Claramunt [36], in their research with adolescents from a community sample of the same country, did not find these results since, although girls scored slightly higher, they did not find significant differences compared to boys. As for the influence of age, although our results could not be compared with other research conducted with adolescents in the child welfare system, it certainly appears that as age increases, so does their knowledge of contraceptive methods. This may be due to an increased interest in safer sex practices with age or simply because visits to sexual and reproductive health services are becoming more frequent with age, especially among girls. Studies with adolescents from a community sample [46] obtain similar results, although they report that the sexuality knowledge of this population remains disturbingly low even in older participants with higher scores. 

The adolescents in our sample also seem to have less positive attitudes towards sexuality, if we compare our results with those obtained by Claramunt [36]. In this regard, previous research with adolescents in the child welfare system reported somewhat contradictory information [8], although in any case, they agreed that it should be an aspect on which to intervene, especially considering that beliefs and attitudes are strongly influenced by family values and experiences, which tend to be particularly negative in these adolescents. This would also justify the findings, which indicate that adolescents in residential care in Eastern Spain show a high internalization of sexist attitudes, much higher than that of young people in the community population [18]. These results support that adolescents who live or have lived in potentially problematic homes are more likely to follow gender roles and develop stereotypical behaviors [13,47]. The findings of our study also confirm that in both males and females’ benevolent sexist manifestations are more frequent than hostile ones [14] as well as in community adolescents [18]. These results reflect the tendency of developed societies to present more covert manifestations of sexism due in part to numerous advances in equality that have occurred in recent decades and that punish the most obvious expressions of sexism [48].

In relation to the latter, these social movements also seem to be bringing about a paradigm shift with respect to the expression of attitudes towards the sexuality of the different sexes. Although traditionally, boys were the ones who showed more positive attitudes towards sexuality [49], currently, both in the community sample [16,36] and in the child welfare system [14], girls seem to present more liberal attitudes towards sexuality and less homo/heterophobic and sexist attitudes. The higher prevalence of sexist attitudes in boys confirms the maintenance of traditional gender roles in adolescents in the child welfare system as a consequence of the influence of patriarchal family contexts [50]. By contrast, age was not shown to be a variable influencing adolescent attitudes, with the exception of hostile sexist attitudes. This is consistent with what has been found in studies with youth in the child welfare system [14], although not in a community sample [18]. The positive relationship between hostile sexism and age could be explained by the greater influence that older adolescents have received from social and cultural models faithful to an even more conservative patriarchal ideology [51].

All of the above is reflected, to some extent, in the sexual behavior of our sample. The vast majority of adolescents in residential care in Eastern Spain interviewed had engaged in sexual activity, with masturbation being the most frequent sexual practice. However, what is most worrying is that they started to engage in sexual exchange activities at a very early age, with almost a quarter of adolescents having their first sexual activity before the age of 13. Other studies with adolescents in the child welfare system have reported similar results [24] and highlighted that their participants engaged in sexual practices much earlier than adolescents in community samples [23]. Being a victim of childhood sexual abuse, suffering intimate partner violence, or presenting mental health and substance abuse problems, has been related to an earlier age of sexual debut [25,52]. Given that these problems are highly prevalent in this population, as well as in our sample (more than 30% had mental health and/or substance abuse problems), this could explain the earlier sexual initiation in adolescents in the child welfare system versus peers in the community sample. However, gender was not shown to be an influential variable either in the age of sexual initiation or in the prevalence of the different sexual practices evaluated, with the exception of masturbation, where boys reported engaging more in this self-pleasure practice. In this sense, it seems that despite the efforts made to encourage the female population to take responsibility for their own body and pleasure, promoting self-exploration and sexual self-stimulation, the results show that the adolescent girls in our sample continue to masturbate considerably less than boys. Genital anatomical, hormonal differences, and, of course, the strong influence of society and culture explain these results [53]. Nevertheless, the absence of differences with respect to the other sexual practices evaluated does reflect some progress in demystifying information about female sexuality. With respect to age, those who report having engaged in sexual practices have a higher mean age than those who have never engaged in these sexual activities. Likewise, older adolescents in our sample also reported later sexual onset.

Despite the high sexual activity of this sample, contraceptive use was low among these adolescents. Less than half of the sample regularly used the male condom, and an even smaller percentage used other contraceptive methods (oral contraceptive pills, hormone patches and injections, or female condoms). Similar results were reported by Cheung et al. [29] in their study of youth in the child welfare system in Texas, with nearly half of their sample using no contraception and one-quarter using condoms. Lambert et al. [25], in their study with adolescents in the child welfare system, also found similar results, although, in our sample, girls presented a slightly higher male condom use prevalence and slightly lower oral contraceptive pills use. Among adolescents in a community sample, although the priority of use of the different methods appears to be similar, the prevalence of use is much greater [54]. It is of concern that withdrawal was cited as the second most used contraceptive method among adolescents in our sample, as it is considered a “non-method,” which does not protect against STIs or unwanted pregnancies. The main reason for engaging in this practice appears to be dissatisfaction with hormonal methods or the desire to express confidence in the partner [55]. It is also worth noting the low use of female condoms compared to male condoms, as in the community sample, due in part to their high cost, lower accessibility, and lack of knowledge regarding their existence [54]. As in other studies of adolescents in the child welfare system [8,25,30], boys appear to make greater regular use of male and female condoms and withdrawal, whereas girls make greater use of oral contraceptive pills and longer-acting contraceptives compared with boys. This shows, once again, the greater vulnerability of girls to contracting STIs [56], but it is also a reflection of the lack of sexual assertiveness they often exhibit, which increases their risk of teen dating violence [57]. Age was also shown to be a significant variable in the use of male condoms and longer-acting contraceptives, with the use of these contraceptive methods becoming more common with the increasing age of the participants. These findings are supported by reports from other studies with youth living in emancipation households [32] and could be explained by cognitive advances with age that influence lower risk-taking [27,28] and greater access to sexual health resources with age.

To our knowledge, this is the first study conducted with adolescents in Spanish residential care focused on exploring their sexual health by analyzing their sexual knowledge, attitudes, and behaviors. However, it is not without limitations. Although the representativeness of the sample and the internal validity have been satisfactory, in order to generalize our results, the number of residential care participants should be increased to increase the external validity and, therefore, its generalization to other contexts. Moreover, the distribution of the sample by gender and age was not completely equal, as there was a higher percentage of boys between 14 and 16 years of age. However, this has been taken into account when analyzing and interpreting the results, and the sample represented the typical gender and age distribution of youth in residential care. Likewise, it should be noted that although all the measurement tools obtained a Cronbach’s Alpha of 0.6 or above in the reliability analysis, some professionals consider this value to be acceptable but not good, so precautions should be taken in this regard. Finally, the existence of a social desirability bias in the participants’ answers should not be ignored since we are dealing with particularly sensitive topics such as sexist attitudes. To control this, the anonymity of the adolescents who volunteered for the study was guaranteed.

## 5. Conclusions

Sexual health is a significant component of overall health, and evidence suggests that for adolescents in the child welfare system, it is even more so, given their life histories marked by experiences of abuse, maltreatment, or neglect. Therefore, this is a critical area that cannot be left unattended if we really want these boys and girls to have healthy development and achieve integral well-being. However, it seems that in this group, on many occasions, no one is assuming the role of a sex educator, which typically corresponds to parents. On the one hand, it is not clear who should assume this responsibility, and on the other hand, resource professionals are not always properly trained and have appropriate educational strategies to implement. Thus, this study should be considered an appeal to the main caregivers of this sample group as well as to those responsible for their training.

This research allows us to conclude that the low level of knowledge about protection methods, the early initiation of sexual exchange activities, and the scarce use of protection methods, together with sexist attitudes, place this group in a situation of great vulnerability. This increases the risk of unwanted pregnancies, sexually transmitted infections, and even dating violence. Child welfare policymakers must also be aware of this reality in order to take action and implement appropriate steps in the area of sex education, not only aimed at adolescents but also at the professionals who care for them and who sometimes do not know how to attend to and manage the needs of the younger people in this area of development. Ensuring greater access to sexual health services at an early age and having spaces in which to share doubts and concerns about this topic can be effective measures, provided they are supported by experts that adolescents can trust.

## Figures and Tables

**Table 1 ijerph-19-12948-t001:** Sexual knowledge and attitudes in Spanish residential care adolescents and differences between age and gender groups.

		Range	Total M (SD)	Age				Gender			
				11–13 M (SD)	14–16 M (SD)	17–19 M (SD)	R	Male M (SD)	Female M (SD)	*t*-Test ^a^	d (CI) ^a^
Knowledge	General	0–17	9.9 (3.2)	8.56 (2.68)	10.16 (3.1)	10.02 (3.4)	0.1	9.24 (2.95)	11.16 (3.31)	5.47 ***	0.62 (0.39, 0.85)
	Contraception methods	0–9	5.16 (1.7)	4.44 (1.48)	5.14 (1.59)	5.44 (1.81)	0.2 ***	4.95 (1.72)	5.57 (1.59)	3.25 **	0.37 (0.14, 0.59)
	STIs	0–9	5.61 (1.93)	4.43 (1.91)	5.88 (1.77)	5.66 (2.01)	0.1	5.32 (1.95)	6.17 (1.77)	3.89 ***	0.45 (0.22, 0.67)
Attitudes	General	0–17	13.36 (2.82)	13.02 (2.63)	13.6 (2.63)	13.19 (3.07)	−0.03	12.81 (2.9)	14.41 (2.34)	5.47 ***	0.59 (0.36, 0.81)
	Ambivalent sexism	20–120	61.46 (21.86)	61.12 (12.09)	59.23 (20.24)	63.79 (24.02)	0.09	67.73 (49.63)	49.63 (19.24)	−7.88 ***	−0.77 (−1.01, −0.54)
	Hostile sexism	10–60	29.81 (11.67)	28.49 (9.73)	28.66 (11.23)	31.37 (12.51)	0.14 *	32.74 (11.58)	24.24 (9.66)	−6.8 ***	−0.85 (1.08, −0.62)
	Benevolent sexism	10–60	31.65 (12.12)	32.63 (11.37)	30.56 (11.42)	32.43 (13.04)	0.04	34.96 (11.27)	25.39 (11.19)	−7.44 ***	−0.89 (−1.13, −0.66

Note: M = Mean; SD = Standard Deviation; r = Pearson correlation; d = standardized mean difference; CI = Confidence Interval. ^a^ Positive scores greater than 0 means favoring females. * *p* < 0.05, ** *p* < 0.01, *** *p* < 0.001.

**Table 2 ijerph-19-12948-t002:** Sexual practices experienced and the age of the first time in Spanish residential care adolescents and differences between age and gender groups.

			Total M (SD)/% (*n*)	Age						Gender			
				11–13 M (SD)/% (*n*)	14–16 M (SD)/% (*n*)	17–19 M (SD)/% (*n*)	M (SD) ^b^	r/*t*-Test	d (CI) ^a^	Male M (SD)/ % (*n*)	Female M (SD)/ % (*n*)	χ^2^/ *t*-Test ^c^	OR (CI)/ d (CI) ^a^
Behavior	Sexual Practices	Masturbation	80.8 (274)	56.1 (23)	80.7 (134)	88.6 (117)	15.95 (1.63)	−4.76 **	−0.72 (−0.99, −0.44)	92 (206)	59.8 (70)	51.44 ***	7.68 (4.19, 14.1)
		Age first time	12.25 (2.31)	10.52 (1.72)	11.79 (2.31)	13.13 (2.07)	NA	0.37 ***	NA	12.27 (2.15)	12.21 (2.71)	−0.19	−0.03 (−0.31, 0.25)
		Mutual masturbation	59.6 (201)	29.3 (12)	63 (104)	64.9 (85)	15.97 (1.57)	−3.09 **	−0.36 (−0.58, −0.14)	56.3 (125)	66.7 (78)	3.42	0.64 (0.4, 1.03)
		Age first time	13.76 (2.01)	10.91 (1.97)	13.2 (1.48)	14.88 (1.89)	NA	0.61 ***	NA	13.98 (2.02)	13.43 (1.93)	−1.84	−0.28 (−0.57, 0.02)
		Oral sex	48.4 (163)	24.4 (10)	47.6 (78)	56.8 (75)	16.1 (1.56)	−3.92 ***	−0.85 (−0.9, −0.6)	47.7 (106)	50.4 (59)	0.22	0.89 (0.57, 1.41)
		Age first time	14.11 (1.98)	11.1 (1.45)	13.53 (1.57)	15.2 (1.69)	NA	0.68 ***	NA	14.28 (2.04)	13.86 (1.86)	−1.24	−0.21 (−0.55, 0.12)
		Vaginal intercourse	71.4 (242)	24.4 (10)	73.9 (122)	82.7 (110)	16.12 (1.47)	−6.23 ***	−0.42 (−0.64, −0.21)	73.7 (165)	67.5 (79)	1.42	1.35 (0.83, 2.19)
		Age first time	13.79 (1.93)	10.67 (1.32)	13.28 (1.61)	14.65 (1.79)	NA	0.49 ***	NA	13.94 (1.99)	13.51 (1.75)	−1.54	−0.22 (−0.51, 0.06)
		Anal sex	18.8 (63)	7.5 (3)	17.1 (28)	24.2 (32)	16.19 (1.44)	−2.67 **	−0.32 (−0.59, −0.05)	19.5 (43)	17.1 (20)	0.28	1.17 (0.65, 2.1)
		Age first time	14.16 (1.8)	11 (1)	13.6 (1.5)	15.04 (1.51)	NA	0.59 ***	NA	14.06 (1.86)	14.35 (1.73)	0.55	0.15 (−0.43, 0.74)

Note: Conditional percentage of the dependent variable knowing the category of the independent variable information. Age first time was used as numerical variable. M = Mean; SD = Standard Deviation; % = frequencies; *n* = number of subjects; r = Pearson correlation; χ^2^ = chi-squared; d = standardized mean difference; OR = Odds Ratios; CI = Confidence Interval; NA = Not Applicable. ^a^ The 95% confidence interval does not include the null value (OR = 1; d = 0); ^b^ Mean age of the group of participants who have engaged in this sexual practice; ^c^ Positive *t*-test scores greater than 0 means favoring females. ** *p* < 0.01, *** *p* < 0.001.

**Table 3 ijerph-19-12948-t003:** Prevalence of Spanish adolescents in residential care who have engaged in sexual activity at 12 years of age or younger, between 13 and 15 years of age, and at 16 years of age or older and gender differences.

		Sexual Practices/Age of First Time	Total			Gender						
						Male			Female			χ^2^
			12 Years Old or Younger % (*n*)	Between 13 and 15 Years Old % (*n*)	16 Years Old or Older % (*n*)	12 Years Old or Younger % (*n*)	Between 13 and 15 Years Old % (*n*)	16 Years Old or Older % (*n*)	12 Years Old or Younger % (*n*)	Between 13 and 15 Years Old % (*n*)	16 Years Old or Older % (*n*)	
Behavior	Sexual Practices	Any sexual activity	46.4 (127)	50.7 (139)	2.9 (8)	47 (86)	51.4 (94)	1.6 (3)	45.1 (41)	49.5 (45)	5.5 (5)	3.19
		Masturbation	44 (109)	51.8 (128)	4 (10)	41.8 (28)	50.7 (34)	7.5 (5)	45 (81)	52.2 (94)	2.8 (5)	2.78
		Mutual masturbation	21.2 (39)	59.3 (109)	19.5 (36)	25.7 (19)	64.9 (48)	9.5 (7)	18.2 (20)	55.5 (61)	26.4 (29)	8.29
		Oral sex	19.6 (28)	56 (77)	23.8 (35)	19.6 (11)	64.3 (36)	16.1 (9)	19.5 (17)	50.6 (44)	29.9 (26)	3.8
		Vaginal intercourse	19.6 (42)	63.7 (137)	16.4 (36)	24.3 (18)	64.9 (48)	10.8 (8)	17 (24)	63.1 (89)	19.9 (28)	3.72
		Anal sex	17.5 (9)	56.8 (29)	25.4 (13)	11.8 (2)	58.8 (10)	29.4 (5)	20.6 (7)	55.9 (19)	23.5 (8)	0.67

Note: Conditional percentage of the dependent variable knowing the category of the independent variable information. % = frequencies; *n* = number of subjects; χ^2^ = chi-squared.

**Table 4 ijerph-19-12948-t004:** Contraceptive methods use in Spanish residential care adolescents and differences between age and gender groups.

			Total % (*n*)	Age						Gender		
				11–13 % (*n*)	14–16 % (*n*)	17–19 % (*n*)	M (SD) ^b^	F-Value	d (CI) ^a, c^	Male % (*n*)	Female % (*n*)	χ^2^
Behavior	Contraception Methods Use	Male Condom						25.74 ***				9.65 **
		Never	35 (116)	82.5 (33)	32.7 (51)	23.3 (31)	14.87 (1.93)			29.6 (64)	45.2 (52)	
		Sometimes	23 (76)	2.5 (1)	28.2 (44)	23.3 (31)	16.07 (1.37)		−0.69 (−0.99, −0.39)	26.9 (58)	15.7 (18)	
		Usually	42 (139)	15 (6)	39.1 (61)	53.4 (71)	16.3 (1.52)		−0.83 (−1.09, −0.57)	43.5 (94)	39.1 (45)	
		Withdrawal						2.04				6.99 *
		Never	64.7 (214)	90 (36)	60.3 (94)	63.2 (84)	15.62 (1.93)			65.3 (141)	63.5 (73)	
		Sometimes	24.2 (80)	10 (4)	26.3 (41)	24.8 (33)	15.9 (146)		−0.15 (−0.41, 0.11)	20.8 (45)	30.4 (35)	
		Usually	11.2 (37)	0 (0)	13.5 (21)	12 (16)	16.19 (1.22)		−0.31 (−0.66, 0.04)	13.9 (30)	6.1 (7)	
		Oral contraceptive pills ^d^						2.11				14.49 **
		Never	78.9 (261)	92.5 (37)	78.2 (122)	75.2 (100)	15.64 (1.8)			83.8 (181)	69.6 (80)	
		Sometimes	13 (43)	7.5 (3)	12.2 (19)	15.8 (21)	16.09 (1.69)		−0.25 (−0.58, 0.07)	12 (26)	14.8 (17)	
		Usually	8.2 (27)	0 (0)	9.6 (15)	9 (12)	16.19 (1.42)		−0.31 (−0.71,0.09)	4.2 (9)	15.7 (18)	
		Female condom ^d^						0.09				0.28
		Never	96.7 (320)	100 (40)	95.9 (149)	97 (129)	15.74 (1.78)			96.3 (208)	97.4 (112)	
		Sometimes	3.3 (11)	0 (0)	4.5 (7)	3 (4)	15.91 (1.04)		−0.096 (−0.69, 0.51)	3.7 (8)	2.6 (3)	
		Usually	0 (0)	0 (0)	0 (0)	0 (0)	15.75 (1.76)		−0.006 (−1.97, 1.96)	0 (0)	0 (0)	
		Hormone patches and injections ^d^						3.14 *				17.34 ***
		Never	86.4 (286)	100 (40)	86.5 (135)	82 (109)	15.65 (1.82)			90.7 (196)	78.3 (90)	
		Sometimes	7.3 (24)	0 (0)	7.1 (11)	9.8 (13)	16.33 (1.17)		−0.38 (−0.79, 0.04)	6.9 (15)	7.8 (9)	
		Usually	6.3 (21)	0 (0)	6.4 (10)	8.3 (11)	16.38 (1.32)		−0.41 (−0.85, 0.04)	2.3 (5)	13.9 (16)	

Note: Conditional percentage of the dependent variable knowing the category of the independent variable information. M = Mean; SD = Standard Deviation; % = frequencies; *n* = number of subjects; χ^2^ = chi-squared; d = standardized mean difference; CI = Confidence Interval. ^a^ The 95% confidence interval does not include the null value (d = 0); ^b^ Mean age of the group of participants who have marked that alternative; ^c^ Standardized mean differences have “Never” as a comparative category; ^d^ The information on the frequency of use of this contraceptive method reported by the men in the sample refers to its use by their partners. * *p* < 0.05, ** *p* < 0.01, *** *p* < 0.001

## Data Availability

The data presented in this study are available on request from the corresponding author.

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
