# Peer review of "Sexual Health among Youth in Residential Care in Spain: Knowledge, Attitudes and Behaviors"

_ijerph, 2022, doi:10.3390/ijerph191912948_

Round 1
Reviewer 1 Report
Thank you very much for give me the opportunity to review the manuscript entitled "Sexual health among youth in residential care in Spain: knowledge, attitudes, and behaviors". I found it very interesting, necessary, del written and well justified. For all these reasons, I think it could be accepted for publication.
As authors said, this is the first study conducted with adolescents in Spanish residential care focused on exploring their sexual health. It is necessary to highlight, therefore, its originality and the relevance of its conclusions for this population and the professionals who work with them, as well as for other researchers in this field.
The theoretical justification is clear, precise, and adequate. It provides an overview of the state of the issue and highlights the gaps in the research, placing the objectives of the study, which are clear and adequate. The Methodology is described clearly and precisely. The analyses carried out are correct, and the results obtained are presented clearly and comprehensibly. The clarity of the tables is appreciated. Finally, the discussion is also accurate and the conclusions perfectly show the implications of the results obtained.
Good job!
Author Response
Thank you very much for your review and your comments. We are very pleased that you found it of interest for publication in this high-quality journal.

Author Response
We were pleased to read about your interest in our research, and of your willingness to consider a revised version of our manuscript for publication in «International Journal of Environmental Research and Public Health». We were also grateful for your constructive and useful comments and recommendations; these comments have been very helpful to improve our paper. We have paid close attention to your concerns, trying to provide the clearest and most appropriate response possible. All changes have been highlighted using track change in order to facilitate further review.
We would be pleased to further discuss our manuscript with you and the reviewers as necessary and hope to hear from you in due course.

Reviewer 3 Report
There are several things in the manuscript to take into account and correct.
1. Reference 35 is a self-citation, which I do not understand why it is placed in one of the instruments, which cannot be consulted nor does it exist yet. Therefore it should not appear.
2. In the abstract, the objective must be stated, as well as a better explanation of the methodology part.
3. In lines 121-122 "To do this, the differences in knowledge, attitudes and behaviors based on the sex and age of the adolescent are analyzed, variables that have been shown to be decisive in samples of community adolescents." It does not correspond to the objectives, it is written as a methodology.
4. In material and method, the type of study must be indicated.
5. From line 130-141 are results. They must be eliminated from the sample/participants. Describe here the sample, the inclusion and exclusion criteria. there should be exclusion criteria, for example by understanding of the language.
6. It is not clear if all the centers and all the minors were authorized.
7. It must be indicated if there is a cut-off point in the instruments and if they are validated in Spanish and by whom. Cronbach's alpha is indicated, but not to which study it belongs.
8. The quality of the research depends on the quality of the instruments, for research according to Nunnally, Cronbach's alpha must be greater than 0.70. Most of these instruments do not have it. (Nunnally JC, Bernstein IH. Psychometric Theory. 3rd New York: McGraw Hill; 1994.)
9. At lines 214-217 "Adolescents had the option of completing the instruments as a semi-structured interview with the professional or autonomously, depending on the participant's comfort as well as their level of comprehension and reading and writing skills too." Indicate why this has not been a bias or if it has been.
10. In row 254, "significant effect" is indicated, in cross-sectional studies one cannot speak of an effect but rather differences or a relationship.
11. All abbreviations in the table must be in notes in each table.
12. In table 2, information is missing, there are not all the data for: M (SD) / % (n)
13. In line 334-335, he refers to studies from other countries, but there is literature in the same country that also studies this knowledge in adolescents. Put this sample better for comparisons since they will have more similar cultural characteristics.
14. In line 340, they comment that it has not been possible to compare with other studies, but there are studies that, although not in the population of centers, study the difference in knowledge and practices in adolescents according to age. Find some study and discuss this part of the manuscript.
15. The conclusions are too long, and the results are repeated. They must be more specific.
Author Response
We were pleased to read about your interest in our research, and of your willingness to consider a revised version of our manuscript for publication in «International Journal of Environmental Research and Public Health». We were also grateful for your constructive and useful comments and recommendations; these comments have been very helpful to improve our paper. We have paid close attention of your concerns, trying to provide the clearest and most appropriate response possible. All changes have been highlighted using track change in order to facilitate further review.
We would be pleased to further discuss our manuscript with you and the reviewers as necessary and hope to hear from you in due course.

Round 2
Reviewer 3 Report
The corrections made have improved the manuscript. Check the format of the tables because some have been misconfigured.